# G-protein-coupled receptors regulate autophagy by ZBTB16-mediated ubiquitination and proteasomal degradation of Atg14L

Tao Zhang[1], Kangyun Dong[1†‡], Wei Liang[1†‡], Daichao Xu[1†‡], Hongguang Xia[2†‡], Jiefei Geng[2], Ayaz Najafov[2], Min Liu[1], Yanxia Li[1], Xiaoran Han[3], Juan Xiao[1], Zhenzhen Jin[4], Ting Peng[4], Yang Gao[4], Yu Cai[1], Chunting Qi[5], Qing Zhang[5], Anyang Sun[1], Marta Lipinski[2§], Hong Zhu[2], Yue Xiong[6], Pier Paolo Pandolfi[7], He Li[4], Qiang Yu[5], Junying Yuan[1,2]*

[1]Interdisciplinary Research Center on Biology and Chemistry, Shanghai Institute of Organic Chemistry, Chinese Academy of Sciences, Shanghai, China; [2]Department of Cell Biology, Harvard Medical School, Boston, United States; [3]Molecular and Cell Biology Lab, Institutes of Biomedical Sciences, Fudan University, Shanghai, China; [4]Department of Histology and Embryology, Tongji Medical College, Huazhong University of Science and Technology, Wuhan, China; [5]Shanghai Institute of Materia Medica, Chinese Academy of Sciences, Shanghai, China; [6]Lineberger Comprehensive Cancer Center, Department of Biochemistry and Biophysics, University of North Carolina at Chapel Hill, Chapel Hill, United States; [7]Cancer Research Institute, Beth Israel Deaconess Cancer Center, Department of Medicine and Pathology, Beth Israel Deaconess Medical Center, Harvard Medical School, Boston, United States

*For correspondence: jyuan@hms.harvard.edu

†These authors contributed equally to this work

‡These authors are co-second authors to this work

Present address: §Department of Anesthesiology, University of Maryland School of Medicine, Baltimore, United States

**Abstract** Autophagy is an important intracellular catabolic mechanism involved in the removal of misfolded proteins. Atg14L, the mammalian ortholog of Atg14 in yeast and a critical regulator of autophagy, mediates the production PtdIns3P to initiate the formation of autophagosomes. However, it is not clear how Atg14L is regulated. In this study, we demonstrate that ubiquitination and degradation of Atg14L is controlled by ZBTB16-Cullin3-Roc1 E3 ubiquitin ligase complex. Furthermore, we show that a wide range of G-protein-coupled receptor (GPCR) ligands and agonists regulate the levels of Atg14L through ZBTB16. In addition, we show that the activation of autophagy by pharmacological inhibition of GPCR reduces the accumulation of misfolded proteins and protects against behavior dysfunction in a mouse model of Huntington's disease. Our study demonstrates a common molecular mechanism by which the activation of GPCRs leads to the suppression of autophagy and a pharmacological strategy to activate autophagy in the CNS for the treatment of neurodegenerative diseases.

## Introduction

Autophagy is an important intracellular catabolic mechanism that mediates the turnover of cytoplasmic constituents via lysosomal degradation. In multi-cellular organisms, autophagy serves important functions in mediating intracellular protein degradation under normal nutritional conditions. Defects in autophagy lead to the accumulation of misfolded proteins in the central nervous system, an organ that is protected from nutritional deprivation under physiological conditions (*Hara et al., 2006*). How cells

**eLife digest** Proteins need to be folded into specific three-dimensional shapes for them to work properly. However, the folding process does not always work perfectly, and proteins are sometimes misfolded. If left to accumulate, these misfolded proteins can damage cells, and most long-term human neurodegenerative diseases, such as Huntington's disease, Parkinson's disease, and Alzheimer's disease, are caused by the build-up of misfolded proteins in the brain.

Autophagy helps to clean up misfolded proteins (and other damaged cell components) by first wrapping them in membrane vesicles. The membrane-wrapped vesicles—known as autophagosomes—then move to fuse with lysosomes, a different kind of membrane compartment in the cell, which breaks down misfolded proteins and recycles the degradation products. In mammalian cells, a protein called Atg14L is critical in the process of autophagosome formation.

The levels of autophagosome formation are regulated by signals that originate from outside the cell. However, it is not clear if and how cells respond to external signals to control the levels of autophagy by regulating the amount of Atg14L. The G-protein-coupled receptors (GPCRs) are the largest class of membrane proteins that our cells have that are involved in sensing and responding to external signals. The activation of GPCRs has been shown to lead to diverse physiological responses. Zhang et al. now show that when any of a wide range of different signaling molecules bind to the GPCRs, the receptors activate a protein called ZBTB16 that leads to the degradation of Atg14L to inhibit autophagy.

Furthermore, Zhang et al. found that blocking the activity of the GPCRs with a drug can activate autophagy and reduce the amount of misfolded proteins in the cell. In mice that have a version of a gene that causes Huntington's disease, this inhibition also protects against the symptoms of the disease. The challenge now is to identify appropriate GPCRs that can be safely manipulated to control the levels of autophagy in the brain in order to reduce the levels of the misfolded proteins that cause neurodegeneration.

regulate autophagy under normal nutritional condition is an important unsolved question in the field. In mammalian cells, adaptor protein Atg14L/Barkor in complex with Vps34, the catalytic subunit of the class III PI3K, and the regulatory proteins Beclin 1 and p150, function as a key driver in orchestrating the formation of autophagosomes by regulating the formation of Vps34 complexes and for targeting to the isolation membrane involved in initiating the formation of autophagosomes (*Obara and Ohsumi, 2011*). However, it remains to be determined how Atg14L is regulated in response to extracellular signaling.

G-protein (heterotrimeric guanine nucleotide–binding protein)-coupled receptors (GPCRs) are important regulators of cellular responses to diverse stimuli with major clinical implications (*Foord et al., 2005*). While the activation of GPCRs is known to lead to numerous downstream events, the role and mechanism of autophagy regulated by GPCRs is not yet clear. Furthermore, it is also not clear how the signaling of GPCRs controls the levels of PtdIns3P.

ZBTB16, also known as promyelocytic leukemia zinc finger or Zfp145, is a member of 'BTB-POZ' protein family and mediates the binding of CUL3, a core component in multiple cullin-RING-based BCR (BTB-CUL3-RBX1) E3 ubiquitin-protein ligase complexes and its substrates (*Furukawa et al., 2003*; *Geyer et al., 2003*; *Xu et al., 2003*). In this study, we investigated the mechanism by which ZBTB16 regulates autophagy. We show that CUL3-ZBTB16 regulates autophagy by mediating the proteasomal degradation of Atg14L, which is controlled by GPCR ligands through GSK3β phosphorylation. Furthermore, we show that inhibiting GPCRs by pharmacological means leads to the activation of autophagy in the central nervous system (CNS) and ameliorates neural dysfunction in a mouse model of Huntington's disease. Our study identified a common mechanism by which multiple GPCR ligands mediate autophagy through regulating the levels of Vps34 complexes and a pharmacological strategy to activate autophagy in the CNS to inhibit neural dysfunction induced by the accumulation of misfolded proteins.

## Results

### Regulation of Atg14L levels by ZBTB16

A genome-wide siRNA screen identified ZBTB16 as one of the hits that when its expression is knocked down can lead to the activation of autophagy and increased production of PtdIns3P (*Lipinski et al., 2010*). Since PtdIns3P is produced by Vps34 complexes, the class III PtdIns3 kinase, we hypothesize that ZBTB16 might affect the levels/activity of Vps34 complexes. Although ZBTB16 is known to be involved in regulation of transcription in nucleus (*Mathew et al., 2012*), ZBTB16 is predominantly localized in the cytoplasm (*Costoya et al., 2008*) (*Figure 1A*), suggesting that ZBTB16 might have a non-nuclear function. To test this hypothesis, we screened for the effects of ZBTB16 knockdown on the levels of Atg14L, Vps34, Beclin1, and UVRAG. Interestingly, we found that knockdown of ZBTB16 led to specific increases in the levels of Atg14L, a key component of Vps34 complexes specifically involved in regulating autophagy, and corresponding increases in the ratio of LC3II/tubulin and a reduction in the levels of p62, indicating the activation of autophagy (*Figure 1B* and *Figure 1—figure supplement 1A*) (*Matsunaga et al., 2009*; *Zhong et al., 2009*). Overexpression of Atg14L has been shown to induce autophagy under normal nutritional conditions (*Matsunaga et al., 2010*; *Fan et al., 2011*). On the other hand, the levels of Beclin1, Vps34, UVRAG were not affected by the knockdown of ZBTB16 (*Figure 1B*). We compared the changes of LC3II/tubulin in the presence or absence of chloroquine (CQ), an inhibitor of lysosomal degradation. We found that compared to that of ZBTB16 knockdown or CQ treatment, the presence of CQ with ZBTB16 knockdown led to a further increase in the ratio of LC3II/tubulin, suggesting that ZBTB16 deficiency led to an increase in autophagic flux (*Figure 1C*).

To confirm the role of ZBTB16 in regulation of Atg14L and autophagy in vivo, we examined the levels of Atg14L and LC3II in ZBTB16−/− mice (*Barna et al., 2000*). Interestingly, we found that the levels of Atg14L were dramatically increased in the kidney, heart, liver, and brain of ZBTB16−/− mice compared to that of wt (*Figure 1D* and *Figure 1—figure supplement 1B*). The levels of autophagy as indicated by the ratio of LC3II/tubulin were significantly higher in ZBTB16−/− mice than that of wt mice. From these results, we conclude that ZBTB16 is an important regulator of Atg14L and autophagy.

To further characterize the impact of ZBTB16 on the levels of Atg14L, we transfected an expression vector of ZBTB16 into HeLa cells. Overexpression of ZBTB16 led to down-regulation of Atg14L protein in a dose-dependent manner but not that of its mRNA (*Figure 1E* and *Figure 1—figure supplement 1C*). Thus, ZBTB16 regulates the protein levels of Atg14L but not its mRNA.

Since ZBTB16 is known to function as an adaptor for CUL3-ROC1 complex of ubiquitin ligase (*Xu et al., 2003*; *Mathew et al., 2012*), we next tested the effect of this complex on the levels of Vps34 complexes. Overexpression of CUL3-ROC1-ZBTB16 led to the reduction in the levels of transfected as well as endogenous Atg14L (*Figure 1F–G*). Overexpression of ZBTB16 also had a minor reducing effect on the levels of Vps34 and Beclin1, but not that of UVRAG (*Figure 1—figure supplement 1D*), consistent with a coordinated regulation of Vps34 complexes (*Itakura et al., 2008*; *Matsunaga et al., 2009*; *Zhong et al., 2009*). The down-regulation of Atg14L by CUL3-ROC1-ZBTB16 can be rescued by MG132, which inhibits the proteasomal degradation, but not by E64d, an inhibitor of lysosomal degradation (*Figure 1F–G*). Finally, we found that knockdown of CUL3 led to increases in the levels of Atg14L and autophagy (*Figure 1—figure supplement 1E*). The effect of knockdown ZBTB16, Cullin3, and ROC1 was also confirmed using image-based LC3-GFP assay (*Figure 1—figure supplement 1F*). Taken together, these results suggest that CUL3-ROC1-ZBTB16 complex may promote the degradation of autophagic-specific Vps34 complexes through proteasomal pathway.

### CUL3-ROC1-ZBTB16 mediates the ubiquitination of Atg14L

Since ZBTB16 is known to function as an adaptor for CUL3-ROC1 ligase complex (*Furukawa et al., 2003*; *Geyer et al., 2003*; *Xu et al., 2003*), we next investigated the possibility that Atg14L might be able to bind to ZBTB16. We found that while the interaction of Atg14L and ZBTB16 was detectable under normal condition, such interaction was dramatically enhanced in the presence of proteasomal inhibitor MG132 (*Figure 2A*; *Figure 2—figure supplement 1A*). In contrast, while the interactions of Vps34 and Beclin1 with ZBTB16 were detectable, the presence of MG132 had only minimum effect (*Figure 2—figure supplement 1B–C*), suggesting that the interaction of ZBTB16 with Vps34 and Beclin1 might be indirect. In addition, we found that the endogenous interaction of ZBTB16 with

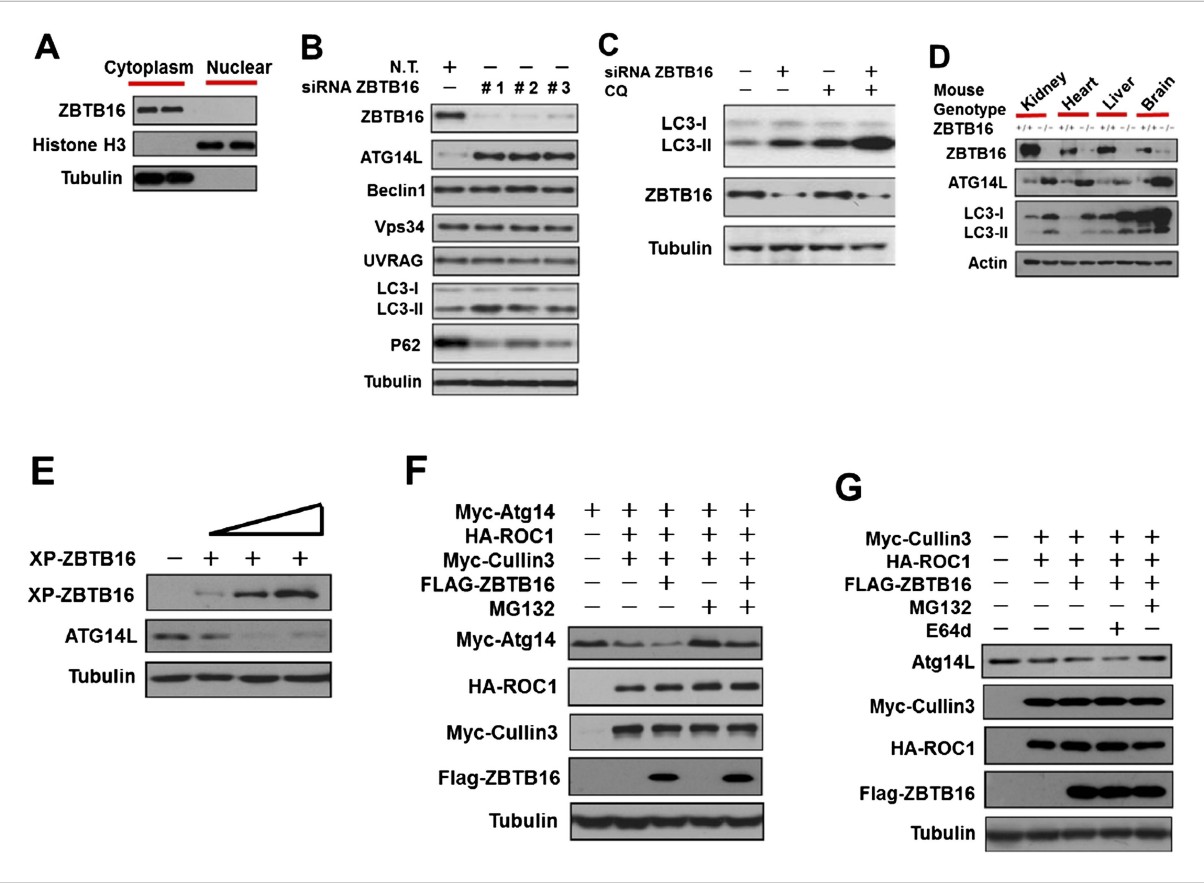

**Figure 1**. ZBTB16 mediates the proteasomal degradation of Atg14L. (**A**) The cytoplasmic and nuclear fractions of HeLa cells cultured in normal media were separated by using Paris Kit (Ambion) and analyzed by western blotting using indicated antibodies. (**B**) HeLa cells were transfected with control siRNA (N.T.) or siRNA targeting ZBTB16 and cultured for 72 hr. The cell lysates were analyzed by western blotting with indicated antibodies. (**C**) HeLa cells were transfected with control siRNA (N.T.) or siRNA targeting ZBTB16 and cultured for 72 hr. Before harvesting, the cells were treated with or without 10 µM CQ (chloroquine) for 4 hr. The cell lysates were analyzed by western blotting with indicated antibodies. (**D**) The levels of ZBTB16, Atg14L, LC3, and actin (control) in the lysates isolated from wt and zbtb16−/− littermates were analyzed by western blotting using indicated antibodies. (**E**) HeLa cells were transfected with the Xpress-tagged ZBTB16 expression vector and cultured for 36 hr. The cell lysate was analyzed by western blotting using anti-Xpress and anti-ATG14L antibodies. Anti-tubulin was used as a loading control. (**F**) 293T cells were transfected with Myc-Atg14, Myc-Cul3, HA-ROC1, and FLAG-ZBTB16 expression vectors and cultured for 24 hr. MG132 (25 µM) was added in the last 8 hr as indicated. The cell lysates were then harvested and analyzed by western blotting using indicated antibodies. (**G**) 293T cells were transfected with Myc-Cul3, HA-ROC1, and FLAG-ZBTB16 expression vectors and cultured for 24 hr. MG132 (25 µM) or E64D (10 µM) was added in the last 8 hr as indicated. The cell lysates were then harvested and analyzed by western blotting using indicated antibodies.

The following figure supplement is available for figure 1:

**Figure supplement 1**. ZBTB16 mediates the proteasomal degradation of Atg14L.

Atg14L can be detected in both 293T and HeLa cells and can also be enhanced in the presence of proteasomal inhibitor MG132 (*Figure 2B–C*). Consistent with the regulation of ZBTB16-Cullin3-ROC1 complex, the interaction of endogenous Atg14L with this complex was also detected (*Figure 2D*).

We next characterized the domains of Atg14L and ZBTB16 involved in their interaction. We found that deletion of the C-terminal Barkor/Atg14(L) autophagosome targeting sequence (BATS) domain, known to be involved in binding of Atg14L to autophagosome membrane (*Fan et al., 2011*), completely eliminated the binding to ZBTB16 (*Figure 2E*). On the other hand, deletion of the center domain of ZBTB16 eliminated the binding to Atg14L (*Figure 2F*). Thus, the C-terminal BATS domain of Atg14L likely interacts with the center domain of ZBTB16.

Consistent with regulation of Atg14L by ZBTB16-mediated ubiquitination, the expression of ZBTB16, Cullin3, and ROC1 led to the ubiquitination of Atg14L (*Figure 2G*). Furthermore, ubiquitination of

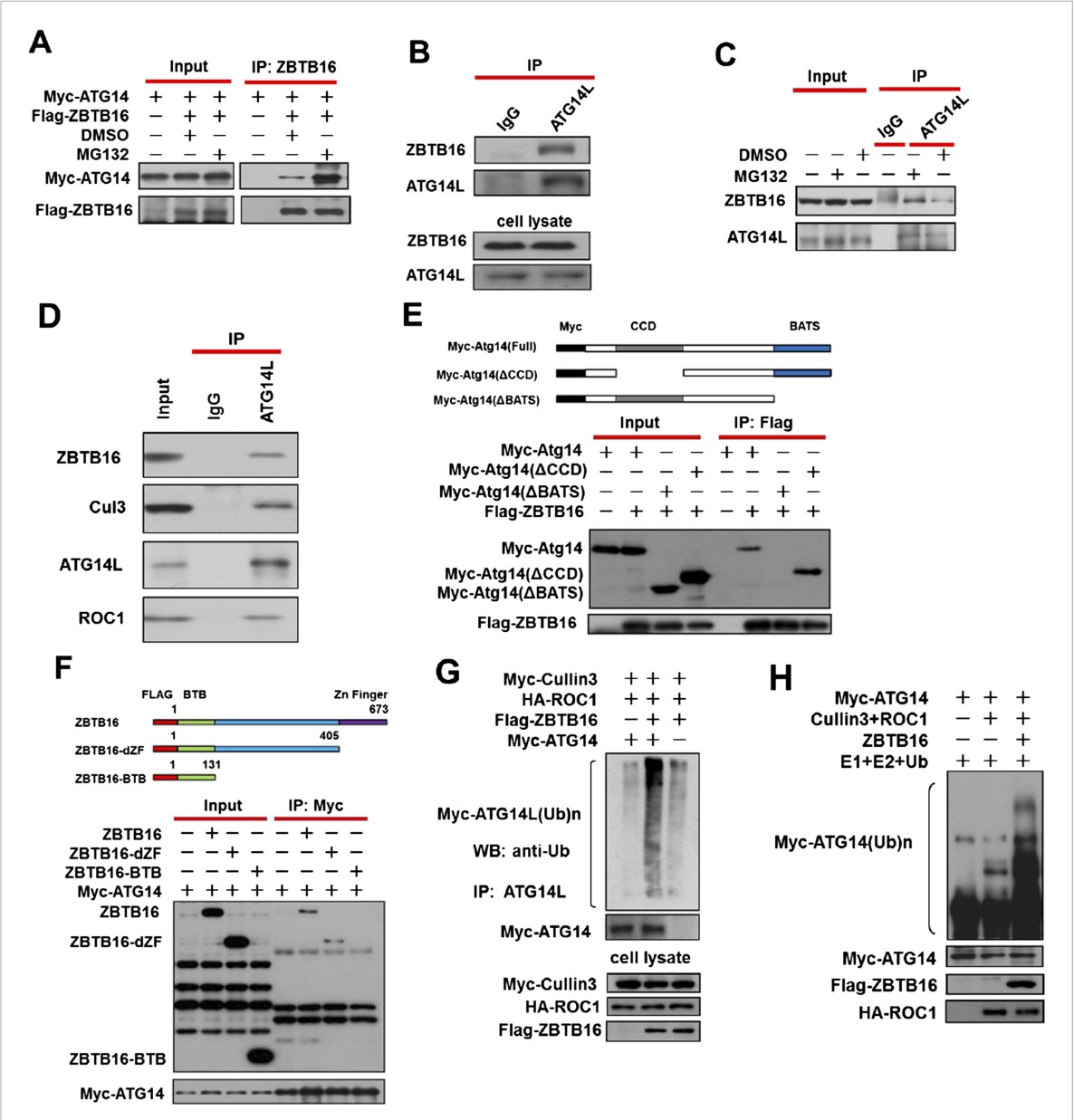

**Figure 2**. Ubiquitination of ATG14L by ZBTB16. (**A**) 293T cells were transfected with expression vectors of FLAG-ZBTB16 and Myc-Atg14 and cultured for 24 hr. The cells were treated with MG132(10 µM) for the last 4 hr before harvesting. The cell lysates were immunoprecipitated with anti-ZBTB16 antibody, and the immunocomplexes were analyzed by western blotting using anti-Myc antibody. (**B**) 293T cells were cultured for 24 hr and then harvested and lysed in Buffer II. The lysates were immunoprecipitated with anti-ATG14L antibody, and the immunocomplexes were analyzed by western blotting using anti-ZBTB16 antibody. (**C**) HeLa cells were treated with MG132(10 µM) for 4 hr and then harvested and lysed in Buffer II. The lysates were immunoprecipitated with anti-ATG14L antibody, and the immunocomplexes were analyzed by western blotting using anti-ZBTB16 antibody. (**D**) HeLa cell lysates were immunoprecipitated with anti-ATG14L antibody, or a control IgG, and the immunocomplexes were analyzed by western blotting using indicated antibodies. (**E**) 293T cells were transfected with the expression vectors of FLAG-ZBTB16, Myc-ATG14L, truncated Myc-Atg14L(ΔBATS), Myc-Atg14L (ΔCCD) as indicated and cultured for 24 hr. The cells were then harvested and lysed in NP-40 buffer. The lysates were immunoprecipitated with anti-Flag antibody, and the immunocomplexes were analyzed by western blotting using anti-Myc antibody. (**F**) 293T cells were transfected with the expression vectors of Myc-ATG14L, FLAG-ZBTB16, truncated FLAG-ZBTB16-dZF, ZBTB16-BTB as indicated and cultured for 24 hr. The cells were then harvested and lysed in NP-40 buffer. The lysates were immunoprecipitated with anti-Myc antibody, and the immunocomplexes were analyzed by western blotting using anti-Flag antibody. (**G**) 293T cells were transfected with expression vectors of Myc-Atg14, HA-ROC1, FLAG-ZBTB16, and Myc-Cul3 expression vectors as indicated and cultured for 24 hr. MG132 (25 µM) was added in the last 6 hr as indicated. The cell lysates were harvested and immunoprecipitated with anti-ATG14L. The immunocomplexes were analyzed by western blotting using anti-Ub antibody for ubiquitin. (**H**) Myc-Cullin3, FLAG-ZBTB16, ROC1, and ATG14L proteins were individually purified from 293T cells transfected with indicated expression vectors by immunoprecipitation. The eluted proteins

*Figure 2. continued on next page*

*Figure 2. Continued*

were incubated with recombinant E1, E2, and Ub. The reactions were terminated by boiling for 5 min in SDS sample buffer. The sample was analyzed by western blotting using anti-Myc.

The following figure supplement is available for figure 2:

**Figure supplement 1**. Ubiquitination of ATG14L by ZBTB16 and Cullin3.

ATG14L can be observed in the presence of purified CUL3, ROC1, and ZBTB16 proteins in vitro (*Figure 2H*). The role of Cullin3 is important as knockdown of CUL3 or expressing CUL3ΔC, a dominant negative mutant (*Jin et al., 2005*; *Mathew et al., 2012*), blocked the ubiquitination of Atg14L, as well as auto-ubiquitination of ZBTB16 (*Figure 2—figure supplement 1D–F*). Taken together, we conclude that CUL3-ROC1-ZBTB16 complex controls the ubiquitination and proteasomal degradation of ATG14L to regulate autophagy.

## Regulation of ZBTB16-mediated ubiquitination of ATG14L

Since regulation of Atg14L and autophagy in cells and mutant mice that are either deficient or overexpressing ZBTB16 occurred under normal nutritional conditions (*Figure 1B–E*), we hypothesized that ZBTB16 might respond to growth factor stimulation, rather than cellular nutritional status. Indeed, we found that serum starvation of HeLa cells for as short as half an hour induced the degradation of ZBTB16 and a concomitant increase in the levels of ATG14L with no effect on its mRNA, while the levels of Beclin1 and Vps34 showed no significant change (*Figure 3A* and *Figure 3—figure supplement 1A*). Similar changes in the levels of Atg14L and ZBTB16 were found in multiple cell lines, including 7721, SK-OV-3, H4, and HCT116 cells, upon serum removal (*Figure 3—figure supplement 1B–E*). On the other hand, no such change in Atg14L or ZBTB16 was observed even after removal of glucose or amino acids (*Figure 3—figure supplement 1F–G*). These results suggest that ZBTB16 and Atg14L are regulated by serum factors rather than the availability of nutrients, such as glucose and amino acids, or the status of mTOR, a major regulator of autophagy in response to nutritional starvation.

The degradation of ZBTB16 in serum-free condition was blocked by the addition of MG132, suggesting serum starvation may lead to increased proteasomal degradation of ZBTB16 (*Figure 3B*). Consistent with this hypothesis, serum starvation also decreased the ubiquitination of ATG14L (*Figure 3C*). However, in HeLa cells deficient for ZBTB16, the levels of Atg14L remained elevated even after the addition of serum (*Figure 3D*; *Figure 3—figure supplement 1H*). Conversely, elevated levels of autophagy as indicated by increased numbers of GFP-LC3 upon serum removal were suppressed after Atg14L knockdown (*Figure 3—figure supplement 1I*). Thus, we conclude that the expression of ZBTB16 is important for the down-regulation of Atg14L levels in the presence of serum.

To examine the potential involvement of class I PI3 kinase signaling in the degradation of ATG14L, we treated HeLa cells with LY294002, an inhibitor of PI3 kinase. As expected, the addition of LY294002 led to the loss of S473 phosphorylation in Akt, a marker for its activation, and the loss of an inhibitory phosphorylation on Ser9 of glycogen synthase kinase-3β (GSK3β) (*Sutherland et al., 1993*), indicating the activation of GSK3β. Interestingly, the treatment with LY294002 led to significant reductions in the levels of ZBTB16 and concomitant increases in the levels of ATG14L and LC3II/tubulin ratio and autophagic flux, suggesting the induction of autophagy (*Figure 3E* and *Figure 3—figure supplement 1J*). Thus, the inhibition of class I PI3 kinase and AKT and activation of GSK3β may contribute to the reduction in the levels of ZBTB16 and the increases that of Atg14L.

Since phosphorylation of substrates provides an important targeting mechanism for Cul3-Roc1 E3 ubiquitin ligase complexes (*Petroski and Deshaies, 2005*), we hypothesized that serum starvation induced degradation of ZBTB16 is regulated by phosphorylation. As inhibition of class I PI3 kinase by LY294002 led to the activation of GSK3β, which is known to be activated by serum starvation to regulate cell survival (*Frame and Cohen, 2001*), we examined the possibility that GSK3β be involved in mediating the degradation of ZBTB16 upon serum starvation. The phosphorylation of Ser9, an inhibitory phosphorylation event, in GSK3β is strongly inhibited in serum- starved HeLa cells as reported (*Figure 3—figure supplement 2A*) (*Sutherland et al., 1993*).

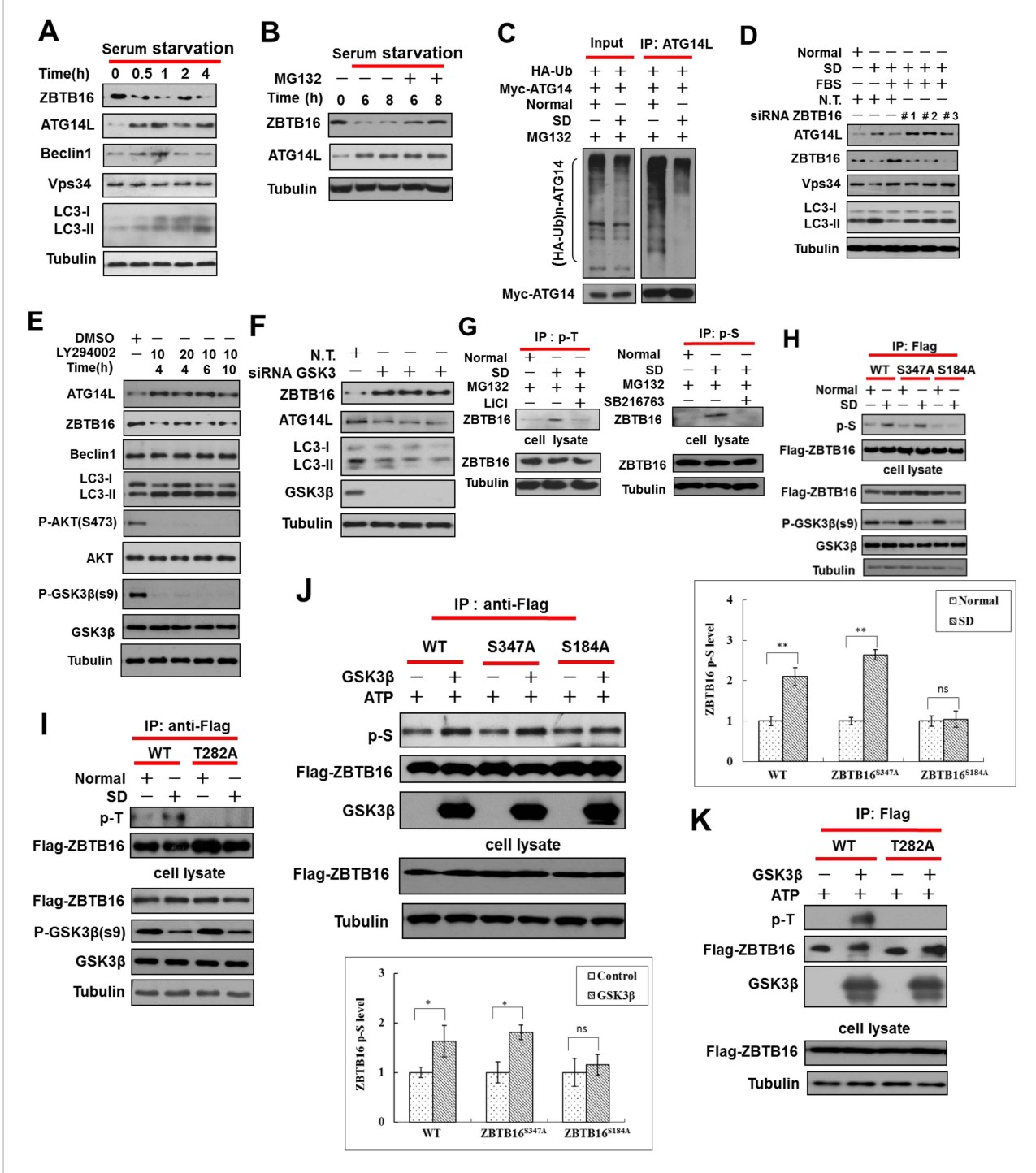

**Figure 3.** Serum starvation regulates ZBTB16 activity through GSK3β. (**A**) HeLa cells were cultured in serum-free condition for indicated periods of time, and then the cell lysates were harvested and analyzed by western blotting using indicated antibodies. (**B**) HeLa cells were serum starved for indicated periods of time with or without MG132 (10 μM), and then the cell lysates were harvested and analyzed by western blotting using indicated antibodies. (**C**) HeLa cells were transfected with the expression vectors of HA-Ub and Myc-ATG14L and cultured for 36 hr and then cultured either in the presence or absence of serum (SD) for an additional 4 hr. The cell lysates were harvested and immunoprecipitated with anti-ATG14L. The immunocomplexes were analyzed by western blotting using anti-HA antibody for ubiquitin or anti-Myc as indicated. (**D**) HeLa cells were transfected with control non-targeting or ZBTB16 targeting siRNA. The cells were cultured in the presence of serum for 46 hr after transfection, subject to serum deprivation (SD) for 24 hr, and then re-stimulated by 10% FBS as indicated for another 2 hr. The total cell lysates were analyzed by western blotting with indicated antibodies. (**E**) HeLa cells were treated with PI3 kinase inhibitor LY294002 at indicated concentrations (μM) and time periods, and the cell lysates were harvested and analyzed by

*Figure 3. continued on next page*

*Figure 3. Continued*

western blotting using indicated antibodies. (**F**) HeLa cells were transfected with control siRNA or siRNA targeting GSK3β and cultured for 72 hr. The total cell lysates were analyzed by western blotting with indicated antibodies. (**G**) HeLa cells were treated with or without SD for 4 hr and then treated with or without MG132 (10 µM), LiCl (10 mM), or SB216763 (20 µM) for an additional 4 hr. The cells were lysed in NP-40 buffer with phosphatase inhibitors and immunoprecipitated with pan-phospho-Thr antibody or pan phospho-Ser antibody. The immunocomplexes were analyzed by western blotting with rabbit anti-ZBTB16. (**H**) The expression vectors of Flag-tagged wild type or ZBTB16 point mutants were transfected into HeLa cells and cultured for 24 hr. Then, the cells were subject to normal or SD conditions as indicated with MG132 (10 µM) for 8 hr. The cells were lysed in NP-40 lysis buffer with phosphatase inhibitors and immunoprecipitated with anti-Flag antibody. The immunocomplexes were analyzed by western blotting with pan phospho-Ser antibody. (**I**) The expression vectors of Flag-tagged wild type or mutants of ZBTB16 were transfected into HeLa cells and cultured for 24 hr. The cells were subject to serum deprived or normal condition with MG132 for 8 hr. Then, the cells were lysed in NP-40 lysis buffer with phosphatase inhibitors and immunoprecipitated with anti-Flag antibody. The immunocomplexes were analyzed by western blotting with pan-phospho-Thr antibody. (**J–K**) The expression vectors of Flag-tagged wild type or mutant ZBTB16 were transfected into 293T cells and cultured for 24 hr. The cell lysates were collected and analyzed by immunoprecipitation with anti-Flag antibody. The immunoprecipitated ZBTB16 was incubated with or without recombinant GSK3β kinase in vitro at 30°C for 1 hr and then analyzed by western blotting analysis with anti-phosphor-Ser (**J**) or anti-phosphor-Thr (**K**). Quantification data are expressed as mean of 3 biological replicates ± SD.

The following figure supplements are available for figure 3:

**Figure supplement 1**. Serum starvation regulates ZBTB16 activity through GSK3β.

**Figure supplement 2**. Serum starvation regulates ZBTB16 activity through GSK3β.

Consistent with a role of GSK3β in regulating ZBTB16, knockdown of GSK3β, or the addition of SB216763, an inhibitor of GSK3β, restored the levels of ZBTB16 (*Figure 3F*; *Figure 3—figure supplement 2B*). Thus, GSK3β might be involved in regulating ZBTB16 in response to serum removal.

We tested the possibility that GSK3β might phosphorylate ZBTB16. Indeed, we found that the expression of constitutively active GSK3β (*Stambolic and Woodgett, 1994*; *Zhao et al., 2012*) could lead to phosphorylation of ZBTB16, which can be inhibited by SB216763 (*Figure 3—figure supplement 2C*). In addition, the interaction of ZBTB16 and GSK3β could be detected, suggesting that ZBTB16 might be a substrate of GSK3β (*Figure 3—figure supplement 2D*). A search of Scansite (http://scansite.mit.edu) identified three possible sites in ZBTB16 for GSK3β: S184, T282, and S347. Interestingly, we could detect increased phospho-ZBTB16 under serum starvation condition, and the phosphorylation of ZBTB16 could be inhibited by inhibitors of GSK3β (*Figure 3G*). We found that S184A mutation, but not S347A mutation, eliminated the phosphorylation of ZBTB16 induced by serum starvation condition detected by pan-phospho-Ser antibody (*Figure 3H*). On the other hand, T282A mutation-eliminated phosphorylation induced by serum starvation condition detected by pan-phospho-Thr antibody (*Figure 3I*). Furthermore, S184A and T282A mutations also eliminated phosphorylation by GSK3β in vitro (*Figure 3J–K*) or in cells expressing constitutively active GSK3 (*Figure 3—figure supplement 2E*) (*Stambolic and Woodgett, 1994*; *Zhao et al., 2012*). Blocking of ZBTB16 phosphorylation, however, had no effect on the mTOR pathway as Torin1, an inhibitor of mTOR, could still induce autophagy in HeLa cells expressing ZBTB16 S184A/T282A (*Figure 3—figure supplement 2F*). Taken together, we conclude that S184 and T282 of ZBTB16 may be phosphorylated by GSK3β in cells under serum starvation condition.

To determine if ubiquitination of ZBTB16 might be affected by serum starvation, we expressed wt and S184A/T282A mutant ZBTB16 in HeLa cells. We found that serum starvation significantly enhanced auto-ubiquitination of wt ZBTB16, but not S184A/T282A mutant (*Figure 4A*). Consistently, auto-ubiquitination of S184D/T282E ZBTB16 mutant was significantly enhanced compared to that of wt (*Figure 4B*). On the other hand, the abilities of S184A/T282A and S184D/T282E ZBTB16 mutants to mediate the ubiquitination of Atg14L were higher and lower than that of wt, respectively (*Figure 4C*). Since both S184 and T282 are localized in the region of ZBTB16 that is important for binding to Atg14L, we tested the effect of phosphor-mimetic mutation in ZBTB16 on the interaction with Atg14L. Consistently, the interaction of S184D/T282E ZBTB16 mutant with Atg14L was significantly weaker than that of wt (*Figure 4D*). Finally, to examine if S184/T282 phosphorylation of ZBTB16 might be critical functionally to regulate autophagy under serum starvation condition, we analyzed the response of HeLa cells expressing wt or S184A/T282A mutant to serum starvation. Since endogenous and overexpressed

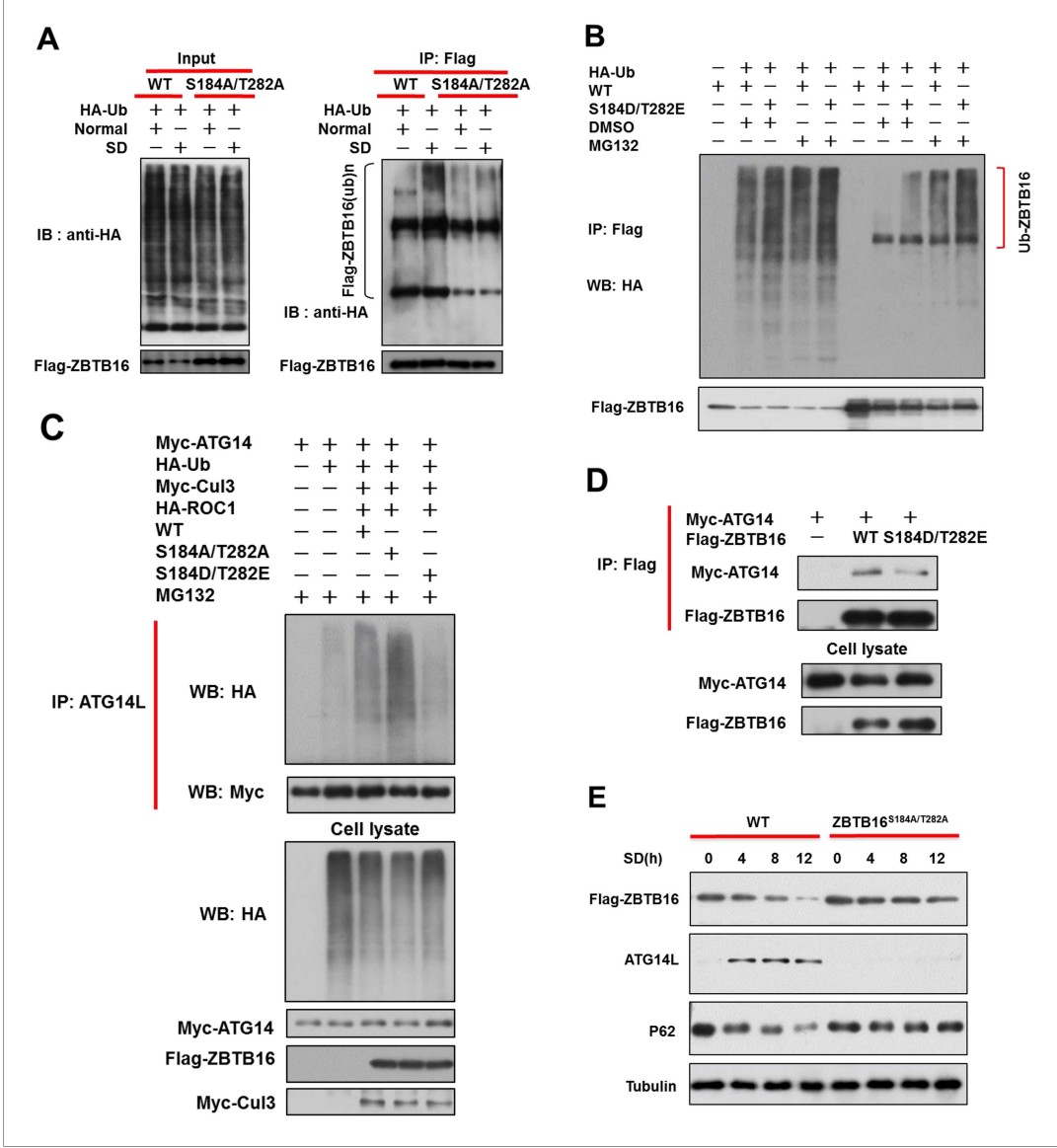

**Figure 4**. Phosphorylation of S184/T282 of ZBTB16 by GSK3b is functionally important for regulating Atg14L under serum starvation condition to promote autophagy. (**A**) The expression vectors of Flag-tagged wild type or mutants ZBTB16 and HA-Ub were transfected into HeLa cells and cultured for 24 hr. HeLa cells were treated with or without serum-deprivation (SD) for 4 hr. The cell lysates were collected and subjected to immunoprecipitation with anti-Flag antibody. The immunocomplexes were analyzed by western blotting using anti-HA for ubiquitin or anti-Flag as indicated. (**B**) 293T cells were transfected with the expression vectors of wide-type FLAG-ZBTB16, mutant FLAG-ZBTB16, HA-Ub as indicated and cultured for 20 hr. The cells were then treated with or without MG132 for 4 hr. Fully denatured lysates were diluted with 0.5% NP-40 lysis buffer and IP with anti-Flag antibody. The lysates were WB with indicated antibodies. (**C**) 293T cells were transfected with the expression vectors of wide-type FLAG-ZBTB16, mutant FLAG-ZBTB16, HA-Ub, Myc-ATG14, HA-ROC1, Myc-Cul3 as indicated and cultured for 36 hr. The cells were then treated with MG132 for 2 hr. Fully denatured lysates were diluted with 0.5% NP-40 lysis buffer and IP with anti-ATG14L antibody. The lysates were WB with indicated antibodies. (**D**) HeLa cells were transfected with expression vectors of wide-type FLAG-ZBTB16, mutant FLAG-ZBTB16, Myc-ATG14 as indicated and then lysed in NP-40 buffer. The lysates were immunoprecipitated with anti-Flag antibody, and the immunocomplexes were analyzed by western blotting using anti-Myc antibody. (**E**) The expression vectors of Flag-tagged wild type or ZBTB16 point mutants were transfected into HeLa cells and cultured for 12 hr. The cells were cultured in serum-free condition for indicated periods of time. The cell lysates were then harvested and analyzed by western blotting using indicated antibodies.

wt ZBTB16 proteins are degraded upon serum starvation while S184A/T282A mutant is not, this experiment provides an opportunity to test if the persistent levels of S184A/T282A mutant under serum starvation condition might be able to suppress the increases in Atg14L and autophagy. Interestingly, we found that the increases in the levels of Atg14L and reduction in that of p62 were blocked by the expression of S184A/T282A (*Figure 4E*). Thus, we conclude that phosphorylation of S184/T282 of ZBTB16 by GSK3β is functionally important for regulating Atg14L under serum starvation condition to promote autophagy.

## Regulation of ATG14L by GPCR signaling

Next, we proceeded to identify the factor(s) in the serum that can reduce the levels of ATG14L. Surprisingly, we found that heat-inactivated fetal bovine serum (FBS) was equally effective in reducing the levels of ATG14L induced by serum starvation (*Figure 5A* and *Figure 5—figure supplement 1A–B*). The presence or absence of serum, however, had no effect on the predominant cytoplasmic localization of ZBTB16 (*Figure 5—figure supplement 1C*). The effect of serum starvation on the increased ATG14L and decreased ZBTB16 levels can also be suppressed by the addition of bovine serum albumin (BSA) with different degree of purity, SDF1, or lysophosphatidic acid (LPA), which are known GPCR ligands (*Figure 5B–D* and *Figure 5—figure supplement 1D–G*). In addition, the treatment of other GPCR ligands such as endothelin 1, high density lipoprotein (HDL) could all block the increases in the levels of Atg14L induced by serum starvation (*Figure 5E* and *Figure 5—figure supplement 1H*). Since the signaling of SDF1 (*Bleul et al., 1996*), LPA (*Hecht et al., 1996*), endothelin (*Horinouchi et al., 2013*), HDL (*Whorton et al., 2007*) is all known to be mediated through GPCRs, these results suggest that the levels of Atg14L are under the control of GPCR signaling.

Heterotrimeric G-proteins, composed of non-identical α, β, and γ subunits, are the critical mediators of GPCR signaling (*Malbon, 2005*). To directly test the role of GPCR signaling on Atg14L, we examined the effect of knocking down multiple Gα proteins. Interestingly, knocking down the expression of multiple Gα proteins led to increases in the levels of ATG14L and autophagic flux (*Figure 5F–G*; *Figure 5—figure supplement 2A–E*). Thus, we conclude that at least a subset of GPCR signaling regulates autophagy through controlling the levels of ATG14L.

The activation of GPCR triggers multiple signal transduction pathways including an increase in the activities of downstream mediators such as Rho-associated coiled-coil containing protein kinase, phospholipases C, which leads to the synthesis of lipid-derived second messengers, activation of adenylyl cyclase, and subsequent activation of protein phosphorylation cascades (*Lappano and Maggiolini, 2011*). Activation of these pathways is known to negatively regulate the activity of GSK3β (*Forde and Dale, 2007*). We next investigated the impacts of these intracellular signaling pathways on Atg14L. Activation of Gαs by a large group of GPCRs stimulates adenylyl cyclase and production of cAMP (*Malbon, 2005*). To directly test the role of cAMP in the induction of Atg14L in serum-starved cells, we treated HeLa cells with forskolin, an activator of adenylyl cyclase, and thereby stimulating the production of cAMP. Interestingly, the treatment of forskolin suppressed the increases of Atg14L under serum starvation condition (*Figure 5H*; *Figure 5—figure supplement 2F*). Conversely, we treated HeLa cells with Pertussis toxin (PTX), which inactivates all members of the Gαi family of G proteins, and found that the treatment of PTX led to reduction of ZBTB16 and increases in Atg14L in the presence of serum (*Figure 5I*; *Figure 5—figure supplement 2G*).

Taken together, these results suggest that ZBTB16 regulated Atg14L degradation is a common downstream mechanism by which multiple signaling pathways that are activated by at least a subset of GPCRs in regulation of autophagy.

## Activation of autophagy by GPCR antagonist promotes autophagy in vivo

AMD3100 is a FDA approved GPCR antagonist used in clinics as an immunostimulant to mobilize hematopoietic stem cells in cancer patients (*Donzella et al., 1998*). The treatment of HeLa cells with AMD3100 led to reduction in the levels of ZBTB16 and increases in the levels of Atg14L (*Figure 6—figure supplement 1A*). Consistent with increases in the levels of Atg14L, AMD3100 treatment also increased the ratio of LC3II and tubulin, which is further stimulated in the presence of CQ, suggesting that AMD3100 promotes autophagic flux (*Figure 6—figure supplement 1B*). Furthermore, the treatment with AMD3100 reduced the levels of mutant Huntington (mHTT), which can

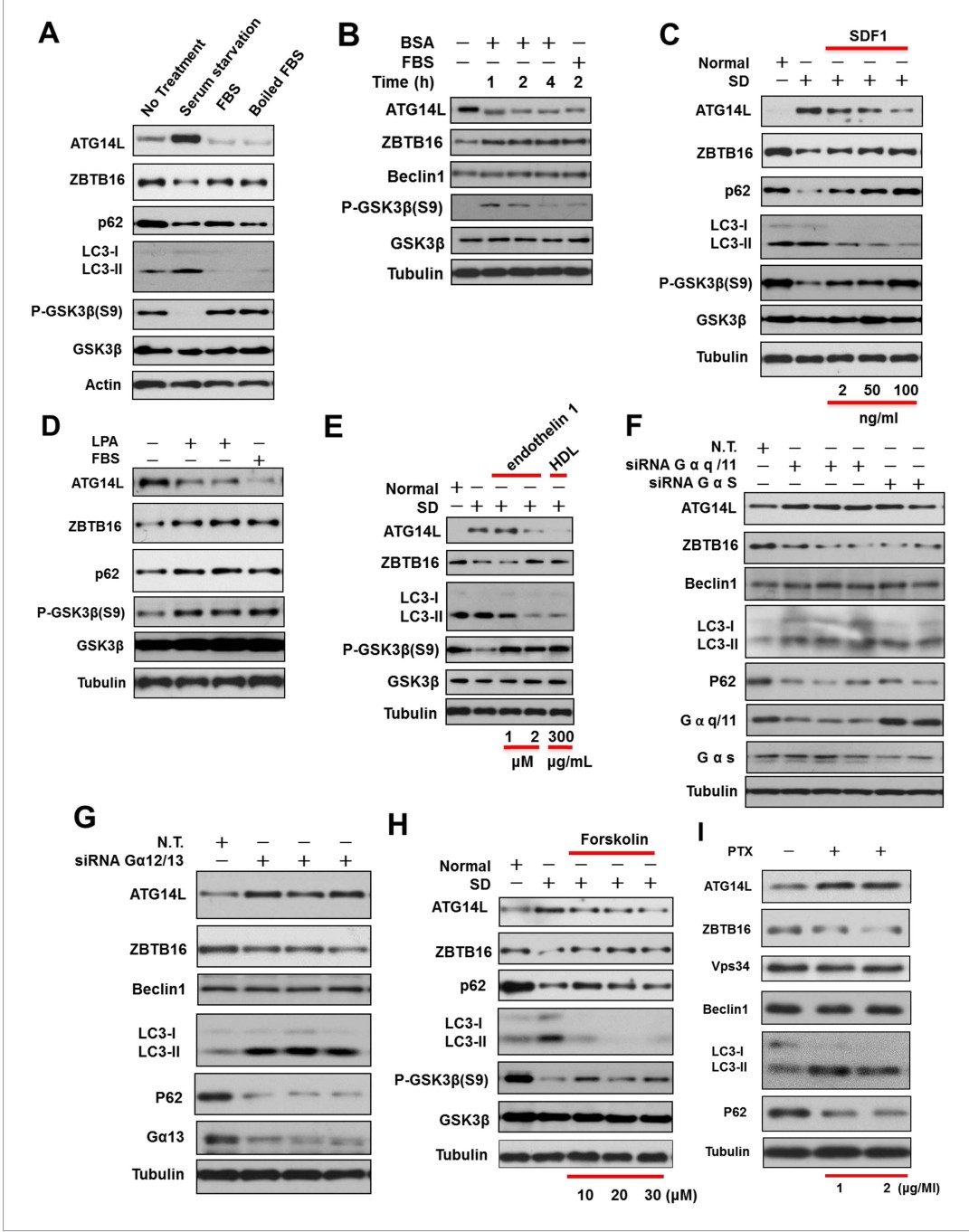

**Figure 5**. Regulation of Atg14L by GPCR mediated signaling pathways. (**A**) HeLa cells were cultured in the presence (no treatment) or absence of serum (serum starvation) for 24 hr and then re-stimulated by 10% FBS or boiled FBS (boiled for 30 min at 95℃) for 1 hr. The cell lysates were analyzed by western blotting with the indicated antibodies. (**B**) Serum-starved HeLa cells were stimulated with 10% FBS or 10 mg/ml BSA (A2058, Sigma) for indicated periods of time and then harvested and analyzed by western blotting using indicated antibodies. (**C**) H4 cells were cultured in the presence or absence of serum for 24 hr. SDF1 was added to serum-starved cells at indicated concentrations for 1 hr. The cells were lysed, and the lysates were analyzed by western blotting with indicated antibodies. (**D**) HeLa cells were serum starved for 24 hr and then stimulated with 20 μM LPA or 10% FBS for 1 hr. The cells were lysed, and the lysates were analyzed by western blotting with indicated antibodies. (**E**) HeLa cells were serum deprived for 24 hr and then stimulated with endothelin 1 or HDL at indicated concentrations for 4 hr. The cells were lysed, and the lysates were analyzed by western blotting with indicated antibodies. (**F**) HeLa cells were transfected with siRNAs for control, Gαq/11, or GαS and cultured for 72 hr. The cell lysates were analyzed by western blotting with indicated
*Figure 5. continued on next page*

*Figure 5. Continued*

antibodies. (**G**) HeLa cells were transfected with siRNAs for control or Gα12/13 and cultured for 72 hr. The cell lysates were analyzed by western blotting with indicated antibodies. (**H**) HeLa cells were serum-deprived (SD) for 24 hr and then stimulated with forskolin for 2 hr. The cell lysates were harvested and analyzed by western blotting using indicated antibodies. (**I**) HeLa cells were treated with Gαi/o inhibitor Pertussis toxin (PTX) at 1 or 2 µg/ml for 6 hr, and the cell lysates were harvested and analyzed by western blotting using indicated antibodies.

The following figure supplements are available for figure 5:

**Figure supplement 1**. Regulation of Atg14L by ligands and agonists of GPCR.

**Figure supplement 2**. Regulation of Atg14L by GPCR mediated signaling pathways.

be blocked by CQ, suggesting that AMD3100 can promote the degradation of mutant Htt through lysosomal-dependent autophagic degradation (*Figure 6—figure supplement 1C*).

We next tested the effect of AMD3100 in mice. We found that in mice dosed with AMD3100, the phosphorylation of AKT(S473) and GSK3(S9) was reduced, and the levels of Atg14L were increased (*Figure 6A*) as that cells under serum starvation in culture. Corresponding to the activation of GSK3β, the levels of ZBTB16 were decreased, while its phosphorylation was increased (*Figure 6A–B*). Autophagy was activated in the brains of these mice treated with AMD3100 as the levels of autophagy marker p62 in the brain were reduced, and ratio of LC3II to tubulin was increased (*Figure 6A*). In addition, we used immunostaining of p62 to measure the levels of autophagy. We found that the levels of p62 in the cortex and striatum of N171-82Q mice treated with vehicle or AMD3100 once daily for one month were significantly lower than that of vehicle-treated mice (*Figure 6—figure supplement 1D*). These data suggest that pharmacological inhibition of GPCR signaling can lead to down-regulation of ZBTB16 and increases in the levels of Atg14L and up-regulation of autophagy in vivo.

To directly examine the impact of activating autophagy by inhibiting GPCR signaling on the accumulation of expanded polyglutamine repeats and disease progression in animal models of expanded polyglutamine disorders, we used N171-82Q transgenic mice, a mouse model of Huntington's disease that expresses the N-terminal 171 amino acids of human htt with 82Q repeat under the control of the prion promoter (*Schilling et al., 1999*). Untreated N171-Q82 mice develop intranuclear expanded polyglutamine inclusions and neuritic aggregates and behavioral abnormalities, including loss of coordination, tremors, hypokinesis, and abnormal gait, before dying prematurely. We found that the treatment of AMD3100 significantly reduced the frequency of hindlimb clasping when suspended by tail (*Figure 6C*). We monitored the effect of AMD3100 on motor function using rotarod test. The motor deficit in N171-Q82 mice is detectable at 10–11 weeks of age (*Schilling et al., 1999*). Beginning at 11 weeks of age, AMD3100 or vehicle control was administered daily to HD mice until the end of their life. When tested at 13.5 weeks of age, the N171-Q82 mice received AMD3100 showed significantly improved motor function as indicated by the extended time on rotarod (*Figure 6D*). The N171-Q82 mice received AMD3100 daily survived significantly longer than that of control group (119 ± 3.82 days vs 104 ± 4.63 days. $p < 0.05$) (*Figure 6E*).

We characterized the accumulation of expanded polyglutamine in vehicle and AMD3100 treated N171-Q82 mice by western blotting and immunostaining. We found that the levels of expanded polyglutamine in brain tissues of N171-82Q mice at the age of 15 weeks were dramatically reduced compared to that of vehicle-treated mice as shown by western blotting (*Figure 6F*). Furthermore, the accumulation of expanded polyglutamine in the cortex, striatum, and hippocampus was significantly reduced after the treatment of AMD3100 as shown by immunostaining (*Figure 6G* and *Figure 6—figure supplement 1E*).

Taken together, we conclude that the activation of autophagy by inhibiting GPCR signaling can promote the degradation of expanded polyQ and preserve neuronal functions by inhibiting Atg14L degradation through ZBTB16-mediated ubiquitination and proteasomal degradation (*Figure 6H*).

## Discussion

Our study reveals a novel mechanism by which GPCRs and the associated intracellular signaling mediators regulate autophagy by controlling the ubiquitination and proteasomal degradation of Atg14L mediated by ZBTB16-CUL3-ROC1. Our study provides a mechanism of crosstalk between

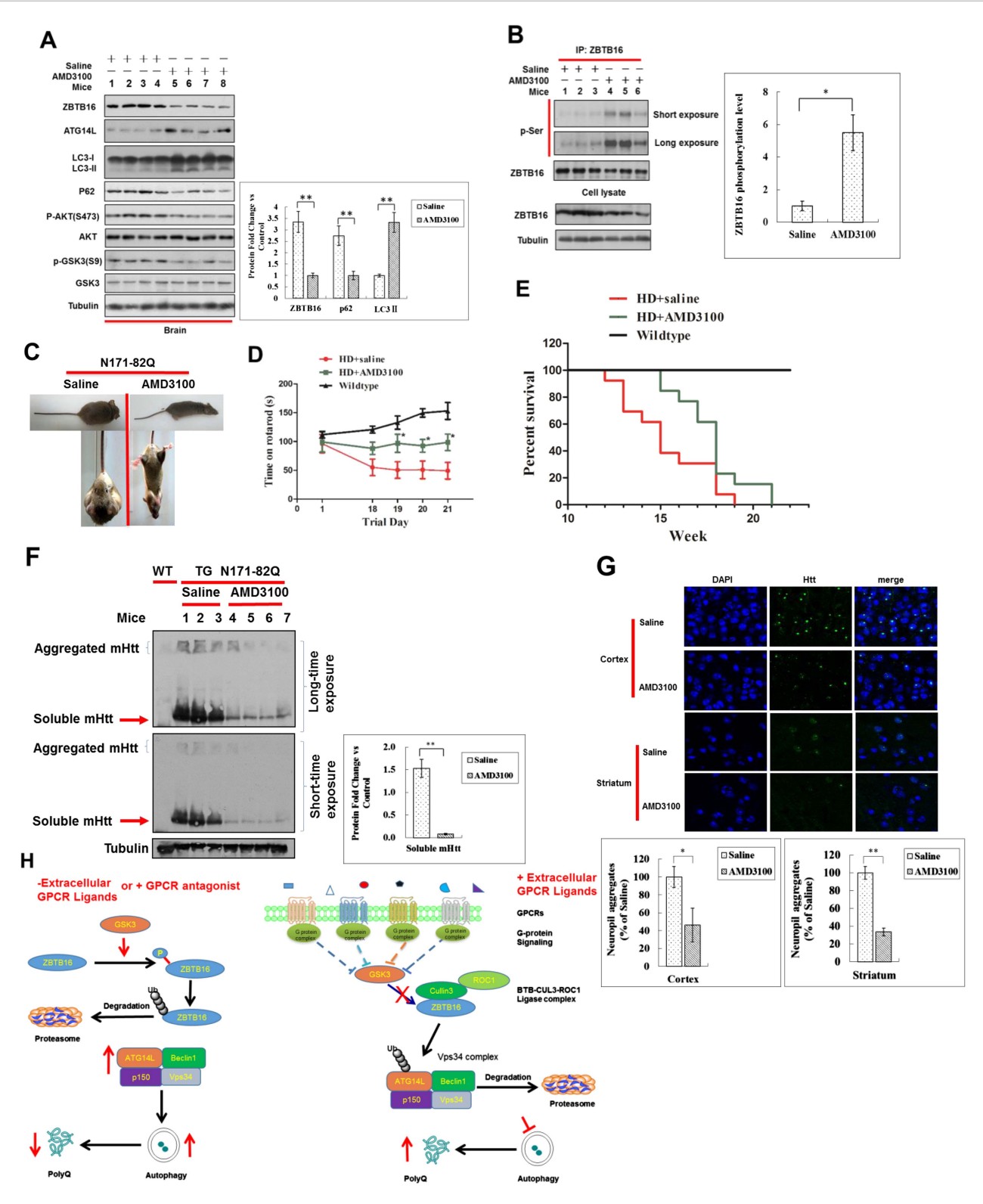

**Figure 6**. Up-regulation of Atg14L and autophagy by GPCR antagonist AMD3100 in vivo ameliorated the neural dysfunction of HD transgenic mice. (**A**) WT mice were dosed once intraperitoneally with AMD3100 at 10 mg/kg body weight and 24 hr later, the brain tissues were collected and analyzed by western blotting using indicated antibodies. Anti-Tubulin was used as a loading control. The levels of ZBTB16, p62, and LC3-II were quantified as graphs on the right side. (**B**) WT mice were dosed once intraperitoneally with AMD3100 at 10 mg/kg body weight and 24 hr later, the brain tissues were lysed in

*Figure 6. continued on next page*

*Figure 6. Continued*

NP-40 buffer with phosphatase inhibitors and immunoprecipitated ZBTB16 antibody. The immunocomplexes were analyzed by western blotting with phospho-Ser antibody and anti-ZBTB16 antibody. The levels of ZBTB16 phosphorylation were quantified shown as a graph on the right. (**C**) Dosing of AMD3100, but not saline alone, for two weeks reduced the clasping of the hindlimbs of N171-82Q mice at 13 weeks of age. (**D**) AMD3100 improved motor function of N171-82Q mice as determined by rotarod testing. The mice received training on rotarod for 10 min each day for 3 days before being tested on the first day of 11 week of age before dosing of AMD3100 started (Day 1). The mice received rotarod training again on day 15–17 during AMD3100 or saline dosing course and were tested again on day 18–21 as shown. For testing, the speed of the rod was set to 5 rpm and increased by 0.5 rpm/s. The data were collected as an average of three trials for each mouse everyday. Mice were allowed to rest for 20 min between trials. N = 13 for each group. The data are presented as mean values ± s.e.m.*. p < 0.05 by t-test for significance. AMD3100 (10 mg/kg body weight) or saline treatment was delivered by intraperitoneal injection from 11 weeks of age daily (every 24 hr). The experiments were carried out in double-blind manner (the person who conducted rotarod testing was unaware if the mouse had received saline or AMD3100, which was provided by another person). (**E**) Survival of wild-type (WT), N171-82Q injected with saline (HD + saline), N171-82Q mice injected with AMD3100 (HD + AMD3100). N = 13 for each group. p < 0.05 by log-rank test for significance. The experiments were carried out in double-blind manner (the person who monitored mouse survival was unaware if the mouse had received saline or AMD3100, which was provided by another person). (**F**) The brain tissues of WT mice or N171-82Q mice dosed for 4 weeks by saline alone or AMD3110 at the age of 15 weeks were isolated and analyzed by western blotting using indicated antibodies. Anti-tubulin was used as a loading control. AMD3100 (10 mg/kg body weight/day) or saline treatment was delivered by intraperitoneal injection from 11 weeks of age every 24 hr. The levels of soluble mHtt were quantified as graphs on the right side. (**G**) Immunostaining of tissue sections from the cortex and striatum of N171-82Q mice treated with saline or AMD3100 daily for 4 weeks using rabbit EM48. DAPI staining is for nucleus. Quantitative assessment of neuropil aggregate density is as graphs below. (**H**) A model for regulation of autophagy under normal nutritional conditions: inhibiting GPCR signaling leads to the activation of GSK3β, which mediates the phosphorylation of ZBTB16 to promote its auto-ubiquitination and inhibit the ubiquitination and proteasomal degradation of Atg14L that in turn leads to increased activity of class III PI3 kinase and production of PI3P, and activation of autophagy.

The following figure supplement is available for figure 6:

**Figure supplement 1**. Up-regulation of Atg14L and autophagy by GPCR antagonist AMD3100 in vivo ameliorated the neural dysfunction of HD transgenic mice.

autophagy and proteasome, two major intracellular degradative machineries. Since GPCRs represent the largest family of membrane-bound receptors that can interact with a broad range of ligands, including small organic compounds, eicosanioids, peptides, and proteins (*Lagerström and Schiöth, 2008*), our results suggest that autophagy can be regulated by a much wider range of extracellular signals and conditions beyond starvation condition than currently appreciated. Furthermore, we demonstrate as a proof-of-principle that suppression of GPCR signaling by AMD3100 can activate autophagy to reduce the accumulation of expanded polyglutamine repeats, extend survival, and ameliorate neurodysfunction in a mouse model of Huntington's disease. Since our study suggests that ZBTB16-regulated Atg14L ubiquitination and degradation is a common downstream mechanism modulated by GPCR signaling in control of autophagy, our study suggests the possibility to explore regulators of additional GPCRs to manipulate autophagy in the CNS. Since GPCRs represent the most common 'druggable' targets, our results demonstrate a potentially wide range of possible targets for safely modulating the levels of autophagy through pharmacological manipulations of GPCR signaling. Our data, however, do not rule out that GPCRs might regulate additional pathways in control of autophagy; nor do we exclude the possibility that ZBTB16 has substrates, in addition to Atg14L, in regulating autophagy.

The phosphorylation of targeting substrates often serves as a signal and prelude for binding, ubiquitination, and proteasomal degradation mediated by Cullin-RING ligase complexes (*Petroski and Deshaies, 2005*). In this case, we found that phosphorylation of S184/T282 in the center domain of ZBTB16 by GSK3β may provide a signal to disrupt the inhibitory folding and thereby activate its auto-ubiquitination activity. On the other hand, the phosphorylation of S184/T282, which is in the domains of ZBTB16 involved in binding to Atg14L, inhibits its interaction with Atg14L and therefore, reduces the ubiquitination and degradation of Atg14L. Since the interaction of ZBTB16 and Atg14L is mediated by the binding to the BATS domain, a lipid-binding domain that binds to highly curved autophagosome structures enriched with PI3P (*Matsunaga et al., 2010*; *Fan et al., 2011*). Consequently, the interaction of ZBTB16 with Atg14L is potentially competitive with the role of Atg14L in promoting the formation of autophagosomes, consistent with its negative role in regulating autophagy. Thus, the degradation of ZBTB16 upon inhibition of GPCR signaling may not only release Atg14L from proteasomal degradation but also provide a permissive signal to allow the binding of the BATS domain with PI3P, an important event during the formation of autophagosomes.

# Materials and methods

### Chemicals and cytokines

From Sigma Aldrich: LY294002, MG132, E64d, SB216763, Forskolin, LPA, endothelin 1, AMD3100, U0126, BSA, Rapamycin, cycloheximide. From Gibco: Pertussis Toxin. Recombinant proteins from Boston Biochem: Ubiquitin Activating Enzyme (UBE1, cat. E-305), UbcH5c (cat.E2-627), Ubiquitin (cat.U-100H). From ProSpec: human SDF1.

### Cell culture

HEK293T, HeLa, human neuroglioma, H4 cells were cultured in Dulbecco's modified eagle medium (DMEM) supplemented with 10% FBS and 1% penicillin-streptomycin (Gibco-BRL). For serum starvation, cells were incubated in DMEM without FBS.

### Transfection

Cells were transfected with plasmid DNA using PolyJet DNA In Vitro Transfection Reagent (Signagen Laboratories, Rockville, MD). siRNA transient transfections were performed using Lipofectamine 2000 (Invitrogen).

### Antibodies

Anti-ATG14L (MBL, PD026,Lot.003), anti-ZBTB16 (ab39354, abcam), anti-Beclin1 (sc-H-300, Santa Cruz), anti-Vps34 (12452-1-AP, PTG), anti LC3B (L7543, Sigma), anti-P62 (PM045, MBL), anti-P-GSK3β (ser9) (#9323, CST), anti-GSK3β (#9832S, CST), anti-p44/42 MAPK kinase (#9102, CST), anti-p-p44/42 MAPK (T202/Y204) (#9101S, CST), anti-pAKT (S473) (#4060S, CST), anti-AKT (sc-H136, Santa Cruz), anti-Tubulin (PM054, MBL), anti-Myc (M4439, Sigma), anti-Flag M2 (F1804, Sigma), anti-Cullin3 (ab75851, abcam), anti-Xpress (R910-25, Invitrogen), anti-phosphoserine (44911M, Invitrogen), anti-p-Threonine-proline (#9391S, CST), Gαq/11 antibody (C-19) (sc-392, Santa Cruz), Gαs antibody (A-16) (sc-26766, Santa Cruz), anti-Gα13 (sc-410, Santa Cruz), anti-Ubiquitin (Z0458, Dako), mouse EM48 (MAB5374, Millipore).

### Immunoprecipitation

Cells were lysed with NP-40 buffer (10 mM Tris-HCl, pH 7.5, 150 mM NaCl, 0.5% NP-40, protease inhibitors cocktail [Sigma], 5% glycerol, 10 mM NaF, 1 mM PMSF), or Buffer II (1 mM EDTA, 0.1% NP-40, 10 mM Tris-HCl pH 7.5). Whole cell lysates obtained by centrifugation were incubated with 5 mg of antibody and protein G agarose beads (Invitrogen) overnight at 4°C. The immunocomplexes were then washed with NP-40 buffer for 3 times and separated by SDS-PAGE for further western blotting assay. Ubiquitination detection was conducted on fully denatured proteins. The cells were lysed with a 1% SDS lysis buffer and boiled for 15 min. Denatured lysates were then diluted with 0.5% NP-40 lysis buffer and immunoprecipitated with appropriate antibody.

### In vitro ubiquitination assay

Myc-Cullin3, FLAG-ZBTB16, HA-ROC1, Myc-ATG14 proteins were immunopurified from 293T cells transfected with the expression vectors encoding above proteins individually. The eluted proteins were incubated with recombinant E1 (60 ng), E2-Ubc5c (300 ng), and ubiquitin (1 μg) in an in vitro ubiquitin ligation reaction containing 50 mM Tris-HCl pH 7.4, 5 mM MgCl$_2$, 10 nM okadaic acid, 2 mM ATP, 0.6 mM Dithiothreitol (DTT), 1 μg ubiquitin (final volume 30 μl). The reactions were incubated at 37°C for 60 min, terminated by boiling for 5 min in SDS sample buffer.

### In vitro GSK3 kinase assay

In vitro kinase reactions were performed using Flag-ZBTB16 proteins immunoprecipitated from HEK293T cells as substrates in kinase reaction buffer (NEB), 200 mM ATP, and 500 units of GSK3 enzyme (NEB p6040S) at 30°C for 1 hr. Reactions were terminated by addition of SDS sample buffer.

### Cytoplasmic and nuclear proteins fractionation

HeLa cells were washed with Phosphate Buffered Saline (PBS) in 4°C and incubated with Buffer A (HEPES [pH 7.9], 20 mM; 10 mM KCl, 1 mM EDTA, 0.5 mM PMSF, 1× protease inhibitor cocktail; 1 mM

DTT, 5 mM NaF, 0.1 mM Na$_3$VO$_4$, 10% Glycerol) at 4°C for 10 min. The cell lysates were collected into cold 1.5-ml EP tube, and NP-40 was added to the final concentration of 0.5%. After a brief vortexing and incubation in ice (1 min), the lysates were centrifuged at 12,500 rpm for 3 min. The supernatant was collected as the cytoplasmic fraction. 50 ml Buffer B (HEPES [pH 7.9], 20 mM; 10 mM KCl, 1 mM EDTA, 0.5 mM PMSF, 1× protease inhibitor cocktail, 1 mM DTT, 5 mM NaF, 0.1 mM Na3 VO4, 420 mM NaCl, 10% Glycerol) was added into to the sediment. The mixture was mixed by vortex and incubated on ice for 40 min before centrifuged at 12,500 rpm for 3 min. The supernatant was collected as the nuclear fraction.

## RT-PCR and site-directed mutagenesis

Total RNA was prepared using RNeasy mini kit (QIAGEN). RNA (1.25 mg) was used for cDNA synthesis using SuperScript First-Strand Synthesis System for Reverse transcription polymerase chain reaction (RT-PCR) (Invitrogen) with oligo dT primers. Mutagenesis was performed using Quik-Change mutagenesis kit (Stratagene). cDNAs for ZBTB16, ATG14L, and their deletion mutants were cloned into pcDNA3.1 using ClonExpressTM II cloning kit (Vazyme biotech). The specific primers were: Vps34, 5′-GAACTTATCCCGTTGCCTTTA-3′ (forward), and 5′-CATGACCTCAGCACTAATCCC-3′ (reverse). Beclin1, 5′-CCGCAAGATAGTGGCAGAAA-3′ (forward), and GCGACCCAGCCTGAAGTTAT-3′ (reverse). Atg14L, 5′-GCTGGTCAACATTCTGTCTCA-3′ (forward), and 5′-CTCCTCAAGGTCTGCTCGTAC-3′ (reverse). Actin, 5′-AGCGAGCATCCCCCAAAGTT-3′ (forward), and 5′-GGGCACGAAGGCTCATCATT-3′ (reverse).

## RNA interference

The sequences of siRNA Oligos used: 5′ to 3′:

Non-target:
UUCUCCGAACGUGUCACGU
ZBTB16 #1
GGGUCGAGCUUCCUGAUAA
ZBTB16 #2
CUAGGGAGCUACACUAUGG
ZBTB16 #3
AGAAGCAUCUGGGCAUCUA
GNAS#1
GCUUGCUUAGAUGUUCCAAAU
GNAS#2
GCCAAGUACUUCAUUCGAGAU
GNAQ
GACACCGAGAAUAUCCGCUUU
GNA11#1
GCUCAAGAUCCUCUACAAGUA
GNA11#2
GCUCAACCUCAAGGAGUACAA
GNA12:
CGUCAACAACAAGCUCUUCUU
GNA13:
GCUCGAGAGAAGCUUCAUAUU
GSK3 β#1
CCCAAAUGUCAAACUACCA
GSK3 β#2
AGUUGGUAGAAAUAAUCAA
GSK3 β#3
GCUAGAUCACUGUAACAUA
Atg14L#1
CCGGGAGAGGUUUAUCGACAAGA
Atg14L#2
AUCUUCGACGAUCCCAUAUAUUA
CUL3#1
GUCGUAGACAGAGGCGCAA
CUL3#2
GAAGGAAUGUUUAGGGAUA

## Animals

N171-82Q mice (B6C3F1/J-Tg(HD82Gln)81Dbo/J, Jackson Laboratory, Bar Harbour, ME) were maintained in the animal facility in accordance with the institutional guidelines.

## AMD3100

Mice received daily intraperitoneal injection of AMD3100 (10 mg/kg of body weight) or saline only. Mice were sacrificed rapidly by cervical dislocation, and the brains were harvested and immediately frozen in liquid nitrogen. Samples were stored at −80°C until use. Frozen tissues were pulverized in liquid nitrogen. All of the subsequent steps were performed at 4°C. Powdered tissue samples were homogenized in 10 vol (wt/vol) of buffer containing 50 mM Tris-HCl pH 7.6, 10 mM EDTA, 100 mM NaF, Protease inhibitor cocktail 1 mM PMSF, 1 mM Na3VO4, 20 mM beta-glycerophosphate, 1 mM sodium pyrophosphate. The samples were then used for western blotting analysis.

## Behavioral analysis

Mice were trained on rotarod for 10 min each day for 3 days before examination. For testing, the speed of the rod was set to 5 rpm and increased by 0.5 rpm/s. Mice received 3 trials daily over 4 consecutive days. Mice were allowed to rest for 20 min between trials. Data from each group were averaged and charted using SPSS software.

## Immunohistochemistry

Mice were anaesthetized with 5% chloral hydrate and perfused intracardially with phosphate-buffered saline (PBS, pH 7.2) followed by 4% paraformaldehyde in PBS at pH 7.2. Brains were removed, cryoprotected in 30% sucrose at 4°C for 3 days, and then sectioned at 10 μm using a cryostat (Leica) at −20°C. The slices were examined by immunofluorescent staining with rabbit EM48 and secondary antibodies conjugated with FITC (Jackson ImmunoResearch Laboratories) or Alexa Fluor 555 (Invitrogen).

## Statistical analysis

Statistical analysis was performed on biological repeats of three independent sets of experiments using ImageJ. The protein levels are determined as a ratio between protein of interest and tubulin. Results were analyzed for statistical significance using two-tailed, unpaired Student's t-test and are expressed as mean ± SD for western blots and mean ± sem for microscopic images. $p < 0.05$ (*); $p < 0.01$ (**); $p < 0.001$ (***). ns, no significance.

## Acknowledgements

We thank Dr Xiaojiang Li for advice on Htt transgenic mice and Htt abs. This work was supported in part by grants (to JY) from the Chinese Academy of Sciences, grants from the National Institute on Aging (US) (R37 AG012859 and R01-AG047231) and the National Institute of Neurological Disorders and Stroke (R01-NS082257) (to JY) and the China National Science Foundation Grant 31129004 (to QY and JY).

## Additional information

### Funding

| Funder | Grant reference | Author |
| --- | --- | --- |
| National Institute on Aging | R37 AG012859 | Junying Yuan |
| National Institute on Aging | R01-AG047231 | Junying Yuan |
| National Natural Science Foundation of China | 31129004 | Qiang Yu, Junying Yuan |
| Chinese Academy of Sciences | | Junying Yuan |

The funders had no role in study design, data collection and interpretation, or the decision to submit the work for publication.

## Author contributions

TZ, ML, QY, JY, Conception and design, Acquisition of data, Analysis and interpretation of data, Drafting or revising the article; KD, WL, DX, HX, XH, ZJ, QZ, HL, Acquisition of data, Analysis and interpretation of data; JG, AN, TP, YG, AS, ML, YX, Conception and design, Analysis and interpretation of data; YL, YC, CQ, PPP, Conception and design, Acquisition of data, Analysis and interpretation of data; JX, Analysis and interpretation of data, Drafting or revising the article; HZ, Conception and design, Acquisition of data

## Ethics

Animal experimentation: All animal experiments were carried out in accordance with an approved protocol by Shanghai Institute of Materia Medica Institutional Animal Care and Use Committee (IACUC). The protocol was approved by the research ethics committee of Chinese Academy of Sciences (Permit Number: 2011-8-YQ-01).

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
