## [Decision Letter]

Thank you for sending your work entitled “GPCRs Regulate Autophagy by ZBTB16-mediated Ubiquitination and Proteasomal Degradation of Atg14L” for consideration at *eLife*. Your article has been favorably evaluated by Vivek Malhotra (Senior editor) and 3 reviewers, one of whom is a member of our Board of Reviewing Editors.

The following individuals responsible for the peer review of your submission have agreed to reveal their identity: Noboru Mizushima (Reviewing editor) and Zhenyu Yue (one of the other two peer reviewers).

The Reviewing editor and the other reviewers discussed their comments before we reached this decision, and the Reviewing editor has assembled the following comments to help you prepare a revised submission.

The authors previously identified ZBTB16 as one of the autophagy-suppressing genes in a genome-wide siRNA screen ([18] Dev. Cell). Zhang et al. show in the present manuscript that the ZBTB16-Cullin3-Roc1 E3 ubiquitin ligase complex induces proteasomal degradation of ATG14L under normal growing conditions. The activity of this E3 complex is inhibited when ZBTB16 is phosphorylated by GSK3b under starvation conditions. Furthermore, the authors show that the GPCR-cAMP pathway negatively regulates the GSK3b, and thereby promotes degradation of ATG14L. Finally, they demonstrate that AMD3100, a GPCR antagonist, increases the level of ATG14L and autophagic flux, and has a therapeutic effect in a Huntington disease mouse model.

Overall, this well-designed, interesting study reveals the novel regulation mechanisms of starvation-induced autophagy and provides an important therapeutic potential of autophagy-regulating drugs. The following points, however, need to be addressed to support and strengthen the overall proposal.

Major comments:

1) While it appears to be clear that the Akt-GSK3b-ZBTB16-ATG14L pathway play a role in starvation-induced autophagy, how much this pathway contributes to the overall autophagy regulation is not very clear. In particular, Akt can also regulate autophagy through inhibition of mTORC1, another important regulator of autophagy, and the major signaling pathway is this unclear. Can inhibition of mTORC1 (e.g. by torin 1) induce autophagy in cells expressing ZBTB16S184A/T282A? Do the levels of ATG14L and ZBTB16 change only by serum starvation, but not by amino acid starvation, which is another potent autophagy-inducible condition? If so, upregulation of ATG14L by ZBTB16 suppression is not critically important for starvation-induced autophagy.

2) It is not shown in this manuscript whether overexpression of ATG14L is sufficient for the increase in autophagic flux. This is a critical point that should be addressed experimentally. Furthermore, it was already suggested that overexpression of Atg14L can induce autophagy even in nutrient-rich conditions ([21] J. Cell Biol., [5] PNAS). The authors should cite and discuss these previous findings.

3) Although the authors show that serum starvation for as short as 30 min induces ZBTB16 reduction and Atg14L accumulation (Figure 3), they determined the induction of autophagy by serum starvation at 24 h (Figure 3). To be consistent, the authors should monitor the LC3 conversion at earlier time points. If autophagy is not induced at earlier time points even though Atg14L accumulation occurs quickly, the authors should clearly mention the fact. The current manuscript may give an impression that Atg14L accumulation quickly induces autophagy and thus may be misleading.

Minor comments:

1) The authors should be careful by not over-interpreting the results. They should include the discussion about the possibility that GPCR-GSK3beta-ZBTB16-Atg14L axis, though important, may represent an over-simplified pathway of autophagy regulation, as GPCR-GSK3beta regulates a variety of pathways and ZBTB16 has many other substrates beyond Atg14L. The observed change in the Atg14L levels alone may not be sufficient to drive alteration of autophagy in particular in the context of AMD3100.

2) It would be helpful for readers if the authors could include a summary model showing how serum starvation or inhibition of GPCRs induces autophagy through stabilization of ATG14L because many factors and negative regulations are involved in the authors' model.

3) As ZBTB16 knockout mice are viable, it would be informative to provide or discuss any phenotypes of these mice that can be explained by a defect in autophagy.

---

## [Author Response]

*1) While it appears to be clear that the Akt-GSK3b-ZBTB16-ATG14L pathway play a role in starvation-induced autophagy, how much this pathway contributes to the overall autophagy regulation is not very clear. In particular, Akt can also regulate autophagy through inhibition of mTORC1, another important regulator of autophagy, and the major signaling pathway is this unclear. Can inhibition of mTORC1 (e.g. by torin 1) induce autophagy in cells expressing ZBTB16S184A/T282A? Do the levels of ATG14L and ZBTB16 change only by serum starvation, but not by amino acid starvation, which is another potent autophagy-inducible condition? If so, upregulation of ATG14L by ZBTB16 suppression is not critically important for starvation-induced autophagy*.

As suggested by the reviewers we have done the assays below, and found that inhibition of mTORC1 by Torin1 could still induce autophagy in HeLa cells expressing ZBTB16 S184A/T282A (Figure 7), and furthermore, amino acid starvation inducing autophagy did not change the levels of ATG14L or ZBTB16 (Figure 7). Figure 7 and B has been incorporated as Figure 3—figure supplement 2 and Figure 3—figure supplement 1, respectively. Thus, we conclude that a subset of GPCRs can suppress autophagy by ZBTB16-mediated ubiquitination and proteasomal degradation independent of mTORC1 or nutritional status.

Author response image 1.A. HeLa cells were transfected with expression vectors of Flag-ZBTB16S184A/T282A and cultured for 24h. Before harvesting the sample, the cells were treated with mTOR inhibitor Torin1(500nM) in nutrient-rich conditions for the indicated periods of time. The cell lysates were analyzed by western blotting with indicated antibodies. B. HeLa cells were cultured in the medium with or without amino acids for the indicated periods of time. The cell lysates were analyzed by western blotting with indicated antibodies.**DOI:**
http://dx.doi.org/10.7554/eLife.06734.018

*2) It is not shown in this manuscript whether overexpression of ATG14L is sufficient for the increase in autophagic flux. This is a critical point that should be addressed experimentally. Furthermore, it was already suggested that overexpression of Atg14L can induce autophagy even in nutrient-rich conditions (*[21]
*J. Cell Biol.,*
[5]
*PNAS). The authors should cite and discuss these previous findings*.

We thank this reviewer for pointing out this key fact that overexpression of Atg14L is sufficient to induce autophagy under normal nutritional condition. We have done a flux experiment and found that overexpression of ATG14L is indeed sufficient for the increase in autophagic flux (Figure 8).

We have specifically cited these two references in this sentence: “Overexpression of Atg14L has been shown to induce autophagy under normal nutritional conditions (5; 21).”

Author response image 2.HeLa cells were transfected with expression vectors of Myc-Atg14 and cultured for 24h. Before harvesting the sample, the cells were treated with or without 10 µM CQ for 4h. The cell lysates were analyzed by western blotting with indicated antibodies.**DOI:**
http://dx.doi.org/10.7554/eLife.06734.019

*3) Although the authors show that serum starvation for as short as 30 min induces ZBTB16 reduction and Atg14L accumulation (*Figure 3*), they determined the induction of autophagy by serum starvation at 24 h (*Figure 3*). To be consistent, the authors should monitor the LC3 conversion at earlier time points. If autophagy is not induced at earlier time points even though Atg14L accumulation occurs quickly, the authors should clearly mention the fact. The current manuscript may give an impression that Atg14L accumulation quickly induces autophagy and thus may be misleading*.

As suggested by the reviewers, we monitored LC3II level in a time course experiment, and found that LC3II level was increased after serum withdrawal at early time points as well (Figure 9). This figure now replaced the original Figure 3.

Author response image 3.HeLa cells were cultured in serum-free condition for indicated periods of time and then the cell lysates were harvested and analyzed by western blotting using indicated antibodies.**DOI:**
http://dx.doi.org/10.7554/eLife.06734.020

Minor comments:

1) The authors should be careful by not over-interpreting the results. They should include the discussion about the possibility that GPCR-GSK3beta-ZBTB16-Atg14L axis, though important, may represent an over-simplified pathway of autophagy regulation, as GPCR-GSK3beta regulates a variety of pathways and ZBTB16 has many other substrates beyond Atg14L. The observed change in the Atg14L levels alone may not be sufficient to drive alteration of autophagy in particular in the context of AMD3100.

We have added one sentence at the end of first paragraph of the Discussion: “Our data, however, do not rule out that GPCRs might regulate additional pathways in control of autophagy, nor do we exclude the possibility that ZBTB16 has substrates, in additional to Atg14L, in regulating autophagy”.

*2) It would be helpful for readers if the authors could include a summary model showing how serum starvation or inhibition of GPCRs induces autophagy through stabilization of ATG14L because many factors and negative regulations are involved in the authors' model*.

Our model is now shown in Figure 6.

*3) As ZBTB16 knockout mice are viable, it would be informative to provide or discuss any phenotypes of these mice that can be explained by a defect in autophagy*.

Male ZTBT16-/- mice are sterile whereas female ZTBT16-/- mice are fertile, suggesting the spermatogenesis is defective without ZBTB16. Autophagy is known to be involved in the process of spermatogenesis. E.g. autophagy has been reported to be involved in paternal mitochondrial destruction during spermatogenesis of Drosophila (Politi et al., 2014). Deletion of Atg7 blocks spermatogenesis at late stage (Wang et al., 2014). Thus, it is possible that the male sterility of ZBTB16 is due to over-activation of autophagy. However, since the male sterility of ZBTB16-/- mice has not been characterized, it is difficult to suggest anything even in the Discussion at this stage.